# Ginsenoside Rb_1_ Reduces Hyper-Vasoconstriction Induced by High Glucose and Endothelial Dysfunction in Rat Aorta

**DOI:** 10.3390/ph16091238

**Published:** 2023-09-01

**Authors:** Jubin Park, You Kyoung Shin, Uihwan Kim, Geun Hee Seol

**Affiliations:** 1Department of Basic Nursing Science, College of Nursing, Korea University, Seoul 02841, Republic of Koreaaniulpo2@korea.ac.kr (U.K.); 2BK21 FOUR Program of Transdisciplinary Major in Learning Health Systems, Graduate School, Korea University, Seoul 02841, Republic of Korea

**Keywords:** ginsenoside Rb1, high glucose, endothelial dysfunction, hyper-vasoconstriction, extracellular Ca^2+^ influx

## Abstract

Acute hyperglycemia induces oxidative damage and inflammation, leading to vascular dysfunction. Ginsenoside Rb_1_ (Rb_1_) is a major component of red ginseng with anti-diabetic, anti-oxidant and anti-inflammatory properties. Here, we investigated the beneficial effects and the underlying mechanisms of Rb_1_ on hypercontraction induced by high glucose (HG) and endothelial dysfunction (ED). The isometric tension of aortic rings was measured by myography. The rings were treated with N^G^-nitro-L-arginine methyl ester (L-NAME) to induce chemical destruction of the endothelium, and Rb_1_ was added after HG induction. The agonist-induced vasoconstriction was significantly higher in the aortic rings treated with L-NAME + HG50 than in those treated with HG50 or L-NAME (*p* = 0.011) alone. Rb_1_ significantly reduced the hypercontraction in the aortic rings treated with L-NAME + HG50 (*p* = 0.004). The ATP-sensitive K^+^ channel (K_ATP_) blocker glibenclamide tended to increase the Rb_1_-associated reduction in the agonist-induced vasoconstriction in the rings treated with L-NAME + HG50. The effect of Rb_1_ in the aortic rings treated with L-NAME + HG50 resulted from a decrease in extracellular Ca^2+^ influx through the receptor-operated Ca^2+^ channel (ROCC, 10^−6^–10^−4^ M CaCl_2_, *p* < 0.001; 10^−3^–2.5 × 10^−3^ M CaCl_2_, *p* = 0.001) and the voltage-gated Ca^2+^ channel (VGCC, 10^−6^ M CaCl_2_, *p* = 0.003; 10^−5^–10^−2^ M CaCl_2_, *p* < 0.001), whereas Rb_1_ did not interfere with Ca^2+^ release from the sarcoplasmic reticulum. In conclusion, we found that Rb_1_ reduced hyper-vasoconstriction induced by HG and ED by inhibiting the ROCC and the VGCC, and possibly by activating the K_ATP_ in rat aorta. This study provides further evidence that Rb_1_ could be developed as a therapeutic target for ED in diabetes.

## 1. Introduction

Pathological conditions can result in acute hyperglycemia. For example, acute pancreatitis can not only impair glucose metabolism [1] but can also induce endothelial injury and dysregulation of vasomotor tone due to inflammatory reactions [2]. High glucose (HG) concentrations have been reported to increase osteogenic protein activity in vascular smooth muscle cells in the presence of elastin-derived peptides and transforming growth factor-beta 1, which can lead to vascular calcifications [3]. In addition, HG was found to significantly increase lysine acetylation and the formation of reactive oxygen species (ROS) in vascular smooth muscle cells, processes that may be associated with impaired vascular smooth muscle cell-dependent vasorelaxation in a murine model of type 2 diabetes mellitus [4]. Hyperglycemia is a major factor in the development of endothelial dysfunction (ED) through oxidative stress and dysregulation of endothelial nitric oxide synthase [5]. A clinical study showed that acute glucose ingestion can increase oxidative stress, possibly degrading vascular endothelial function, even in healthy young men [6]. Blood sugar control alone did not effectively prevent cardiovascular complications in patients with type 2 diabetes and hyperglycemia [7]. Consequently, in patients with acute hyperglycemia, efforts should be made to prevent and treat abnormally increased vascular tone under conditions of ED.

Red ginseng (Ginseng Radix Rubra), a plant belonging to the family *Araliaceae* [8], has been used in traditional Korean medicine due to the pharmacological effects of the ginsenoside saponins contained in its roots, such as anti-oxidant, anti-inflammatory, and anti-diabetic properties [9]. It has been reported that there are several types of ginsenosides in red ginseng [10]. Ginsenosides can be divided into protopanaxadiol-type compounds, including Rb_1_, Rb_2_, R_c_, and R_d_, and protopanaxatriol-type compounds, including Rg_1_, Rg_2_, and Rh_1_, with both of these types possessing a dammarane triterpenoid structure [11]. The ginsenosides Rb1 (hereinafter, Rb_1_) and Rg1 (hereinafter, Rg_1_) used in the present study are representative components of protopanaxadiol-type and protopanaxatriol-type, respectively [12]. The chemical structures of Rb_1_ and Rg_1_ differ in aspects such as the positions of sugar moieties [13]. Both Rb_1_ and Rg_1_ are known to have antidiabetic effects [14,15]. Especially, Rb_1_ accounts for the largest proportion of ginsenoside saponins contained in red ginseng [10,16]. Rb_1_ was shown to stimulate the production of nitric oxide in vascular endothelial cells [17]. Rb_1_ was also found to down-regulate the Wnt/β-catenin pathway, which is associated with vascular calcification in rat vascular smooth muscle cells [18]. In pulmonary arteries, Rb_1_ was reported to inhibit store-operated Ca^2+^ entry (SOCE) by suppressing stromal interaction molecule activation rather than by altering Ca^2+^ release from the sarcoplasmic reticulum (SR), and thereby to decrease endothelin-1-induced vasoconstriction [19]. Furthermore, daily intraperitoneal injection of Rb_1_ for three weeks was reported to decrease SOCE and vasoconstriction in the pulmonary arteries of rats with pulmonary hypertension [20]. However, the effects of Rb_1_ on vasoconstriction under hyperglycemia and ED conditions remained unclear. Therefore, in the present study, we investigated the effects of Rb_1_ on hyper-vasoconstriction induced by HG and ED in rat aorta, and its underlying mechanisms.

## 2. Results

### 2.1. Effects of HG on Vasoconstriction in Rat Aorta

Analyses of the effects of HG on the agonist-induced vasoconstriction of aortic rings showed that HG concentrations of 25 mM (HG25) or 50 mM (HG50) did not significantly affect the agonist-induced vasoconstriction of rings with intact endothelium (Control, 1.08 ± 0.04 g; HG25, 1.05 ± 0.08 g; HG50, 1.13 ± 0.08 g). Also, there was no significant difference in the agonist-induced vasoconstriction between the rings treated with N^G^-nitro-L-arginine methyl ester (L-NAME, 1.10 ± 0.06 g) and L-NAME + HG25 (1.18 ± 0.06 g). In contrast, the agonist-induced vasoconstriction was significantly higher in the rings treated with L-NAME + HG50 (1.32 ± 0.06 g) than in the control (*p* = 0.007) and L-NAME-treated (*p* = 0.011) rings. The agonist-induced vasoconstriction of aortic rings was not significantly different between the L-NAME + Mannitol (1.22 ± 0.03 g) group and the control group. These results suggest that vasoconstriction was significantly increased only after simultaneous treatment with HG50 and L-NAME for denudation of the endothelium, and thus was attributable to the HG condition rather than a hyperosmotic effect (Figure 1A).

### 2.2. Effects of Rb_1_ on Hyper-Vasoconstriction Induced by HG and ED in Rat Aorta

The effects of 10^−6^ or 10^−5^ M Rb_1_ (called Rb_1_ 1 and Rb_1_ 10, respectively) or Rg_1_ (called Rg_1_ 1 or Rg_1_ 10, respectively), or 10^−5^ M N-acetylcysteine (NAC 10) on hyper-vasoconstriction induced by HG and ED were evaluated. The agonist-induced vasoconstriction in the aortic rings treated with L-NAME + Rb_1_ 1 (1.09 ± 0.04 g), L-NAME + Rb_1_ 10 (1.11 ± 0.05 g), and L-NAME + Rg_1_ 10 (1.14 ± 0.05 g) did not differ significantly from the agonist-induced vasoconstriction in the rings treated with L-NAME alone (1.10 ± 0.06 g). Similarly, the agonist-induced vasoconstriction in the aortic rings treated with L-NAME + HG25 + Rb_1_ 1 (1.18 ± 0.04 g), L-NAME + HG25 + Rb_1_ 10 (1.11 ± 0.05 g), and L-NAME + HG25 + Rg_1_ 10 (1.19 ± 0.04 g) did not differ significantly from the agonist-induced vasoconstriction in the aortic rings treated with L-NAME + HG25 (1.18 ± 0.06 g). Compared to the aortic rings treated with L-NAME + HG50 (1.32 ± 0.06 g), the agonist-induced vasoconstriction in the rings treated with L-NAME + HG50 + Rb_1_ 10 (1.11 ± 0.04 g, *p* = 0.004) was significantly reduced to a level similar to that seen in the L-NAME + HG50 + NAC 10 (1.14 ± 0.06 g) group; however, such a reduction was not seen in the rings treated with L-NAME + HG50 + Rg_1_ 10 (1.28 ± 0.05 g) (Figure 1B).

### 2.3. Effect of K^+^ Channel Blockers on Rb_1_-Treated Hyper-Vasoconstriction Induced by HG and ED in Rat Aorta

To assess the mechanism underlying the effect of Rb_1_ 10 on agonist-induced vasoconstriction, aortic rings were treated with several K^+^ channel blockers. Tetraethylamine (TEA, 1.09 ± 0.04 g), barium chloride (BaCl_2_, 1.11 ± 0.03 g), and 4-aminopyridine (4-AP, 1.09 ± 0.04 g) did not significantly alter the agonist-induced vasoconstriction in the aortic rings treated with L-NAME + HG50 + Rb_1_ 10 (1.11 ± 0.05 g), whereas the agonist-induced vasoconstriction tended to be higher in the rings treated with L-NAME + HG50 + Rb_1_ 10 + Glibenclamide (Gliben, 1.21 ± 0.03 g) than with L-NAME + HG50 + Rb_1_ 10 (1.11 ± 0.05 g; *p* = 0.062). These results suggested that activation of the ATP-sensitive K^+^ channel (K_ATP_) may be involved in the inhibitory effect of Rb_1_ on hyper-vasoconstriction induced by HG and ED (Figure 2A).

### 2.4. Effect of SR Ca^2+^ Release and Extracellular Ca^2+^ Influx on Rb_1_-Treated Hyper-Vasoconstriction Induced by HG and ED in Rat Aorta

To further explore the mechanism of the action of Rb_1_, its effects on vasoconstriction induced by SR Ca^2+^ release and extracellular Ca^2+^ influx were analyzed. Rb_1_ 10 did not significantly affect vasoconstriction due to SR Ca^2+^ release, as shown by comparing the aortic rings treated with L-NAME + HG50 (0.30 ± 0.02 g) and L-NAME + HG50 + Rb_1_ 10 (0.32 ± 0.02 g) (Figure 2B). In contrast, Rb_1_ 10 significantly reduced vasoconstriction due to extracellular Ca^2+^ influx via the receptor-operated Ca^2+^ channel (ROCC, 10^−6^–10^−4^ M, *p* < 0.001; 10^−3^–2.5 × 10^−3^ M, *p* = 0.001) (Figure 2C) and the voltage-gated Ca^2+^ channel (VGCC, 10^−6^ M, *p* = 0.003; 10^−5^–10^−2^ M, *p* < 0.001) (Figure 2D).

## 3. Discussion

The present study found that the agonist-induced vasoconstriction of aortic rings with intact endothelium was not affected by the HG concentration. In the presence of ED, however, the agonist-induced vasoconstriction was significantly increased by the addition of 50 mM glucose. This concentration is relevant to that seen in the hyperosmolar hyperglycemic state [21] leading to acute pancreatitis [22]. This effect was independent of vascular contracting factors from the endothelium, such as endothelin-1 [23], indicating that HG has a direct effect on vascular smooth muscle cells. Furthermore, a mannitol-based hyperosmotic control did not show a vasoconstriction increase comparable to that seen in the HG50 group, suggesting that the HG was responsible for enhancing the agonist-induced vasoconstriction of vascular smooth muscle cells. The hyperpolarization of mitochondria in mouse vascular smooth muscle cells treated with HG was found to inhibit myosin light chain phosphatase, resulting in vascular smooth muscle contraction [24]. The exposure of murine aortic vascular smooth muscle cells to 30 mM glucose for 48 h was found to increase intracellular Ca^2+^ levels through SOCE [25]. Because acute pancreatitis is frequently accompanied by ED [2] and hyperglycemia [26], intracellular Ca^2+^ concentrations may be increased in the vascular smooth muscle cells of patients with acute pancreatitis, resulting in vascular hypercontraction.

The ability of Rb_1_ and Rg_1_ to reduce hyper-vasoconstriction induced by HG and ED was also evaluated. Although neither Rb_1_ nor Rg_1_ altered vasoconstriction in the aortic rings treated with L-NAME and L-NAME+HG25, Rb_1_ significantly reduced hyper-vasoconstriction induced by 50 mM glucose and ED, whereas Rg_1_ was ineffective. Rb_1_ was more effective than Rg_1_ in mitigating vascular smooth muscle dysfunction in the presence of angiotensin 2-induced abdominal aortic aneurysm [27] and inhibiting vascular inflammatory action in the coronary artery endothelium [28]. These different effects of Rb_1_ and Rg_1_ may reflect differences in their chemical configurations, such as the positions of sugar moieties and aglycone structures [13].

The effects of Rb_1_ on hyper-vasoconstriction induced by HG and ED were not altered by the K^+^ channel inhibitors, TEA, BaCl_2_, and 4-AP. In contrast, the K_ATP_ blocker Gliben tended to suppress the Rb_1_ inhibition of hyper-vasoconstriction induced by HG and ED. HG has been found to inhibit K_ATP_ activity in vascular smooth muscle, an inhibition mediated by increased superoxide production in the human omental artery [29]. Furthermore, the K_ATP_ is inhibited when intracellular ATP is decreased, followed by the opening of the VGCC, resulting in increased cytosolic Ca^2+^ levels [30,31]. Thus, Rb_1_ may inhibit extracellular Ca^2+^ influx through the activation of the K_ATP_, thereby reducing hyper-vasoconstriction induced by HG and ED.

Rb_1_ has been found to reduce intracellular Ca^2+^ concentrations in the myocardial H9C2 cell hypoxia model, through a mechanism involving the downregulation of the calcium/calmodulin-dependent protein kinase II and the ryanodine receptor 2 [32]. Furthermore, Rb_1_ selectively suppressed the L-type VGCC in cultured rat hippocampus neurons [33]. To determine whether the effects of Rb_1_ on hyper-vasoconstriction induced by HG and ED were associated with Ca^2+^ flow, the effects of Rb_1_ on vasoconstriction induced by sarcoplasmic SR Ca^2+^ release or extracellular Ca^2+^ influx were measured. The ability of Rb_1_ to ameliorate the increased vasoconstriction induced by treatment with L-NAME and 50 mM glucose was found to be independent of SR Ca^2+^ release. In contrast, Rb_1_ was found to reduce hyper-vasoconstriction induced by HG and ED through inhibiting extracellular Ca^2+^ influx via the ROCC and the VGCC, regarded as the main Ca^2+^ channels for controlling vascular tension in vascular smooth muscle [34,35].

Pretreatment with Rb_1_ was found to inhibit Ca^2+^ increase through SOCE in pulmonary arterial smooth muscle cells [20]. Moreover, Rb_1_ significantly inhibited Ca^2+^ influx through SOCE only in vascular smooth muscle cells, but not in vascular endothelial cells, exposed to 30 mM glucose for 48 h [25]. In agreement with these findings, the present study found that Rb_1_ did not affect the agonist-induced vasoconstriction in aortas with HG and intact endothelium. This lack of effect may be due to endothelium-derived contracting factors, including angiotensin II [36], thromboxane [37], and endothelin-1 [38], released by HG stimulation.

Rb_1_ is expected to have therapeutic potential in diseases accompanied by hyperglycemia and ED. The present study indicates that Rb_1_, which possesses anti-diabetic properties [39], reduced hyper-vasoconstriction induced by HG and ED through the ROCC- and VGCC-mediated inhibition of extracellular Ca^2+^ influx and/or possibly through the K_ATP_ channel activation.

These findings indicate the importance of maintaining healthy endothelium to prevent hypercontraction of vascular smooth muscles, especially in patients with hyperglycemia. If endothelial vessels are damaged, however, it is crucial to manage hypercontraction of vascular smooth muscle in hyperglycemic conditions. The finding that Rb_1_ is effective as a preventive and therapeutic intervention for hyper-vasoconstriction induced by HG and ED in rat aorta provides a potential rationale for clinical trials aimed at evaluating the efficacy of Rb_1_ in patients with both hyperglycemia and ED.

## 4. Materials and Methods

### 4.1. Animals

Three-week-old male Sprague Dawley rats (weighing 65 ± 15 g) were purchased from Young Bio (Seongnam, Korea) and housed at a temperature of 21–23 °C and under a 12 h photoperiod. The rats were given free access to tap water and a standard chow diet comprised of 60% carbohydrate, 27% protein, and 13% fat (percentages of total kcal). After acclimatization, the rats were assigned to distinct groups (*N* = 4–11). The experimental protocol in this study was approved by the Institutional Committee for Animal Research Ethics of Korea University (KUIACUC-2021-0088).

### 4.2. Preparation of Aortic Rings

Rats were anesthetized with isoflurane, and their thoracic aortas were dissected. The aortic tissue was immersed in Krebs solution (118.3 mM NaCl, 25 mM NaHCO_3_, 1.22 mM KH_2_PO_4_, 11.1 mM glucose, 4.78 mM KCl, 1.2 mM MgCl_2_, 2.5 mM CaCl_2_) and the connective tissue was removed carefully. Each aorta was cut into rings 2–3 mm in length and placed in 37 °C chambers, which were continually aerated with a mixture of 5% CO_2_ and 95% O_2_. The aortic rings were connected to a DMT 620M (Danish Myo Technology, Aarhus, Denmark) with tungsten wires, and then stabilized at 1.0 g tension for 80 min. The aortic rings were pretreated with 10^−4^ M L-NAME to denudate the endothelium chemically and thus to assess only the vasoconstrictive responses of vascular smooth muscle. For the HG condition, the total glucose concentration in each chamber was adjusted to 25 mM [40] or 50 mM [41] for 30 min. Mannitol was used to prepare a high osmolarity control corresponding to HG50. The aortic rings were pretreated with 10^−6^ or 10^−5^ M Rb_1_ or Rg_1_, or with 10^−5^ M NAC, (an antioxidant) [42], and contraction was induced by adding 10^−5^ M PE. When the maximum vasoconstriction plateau was reached, 10^−5^ M acetylcholine was added to confirm endothelial denudation.

### 4.3. Involvement of K^+^ Channel in Vasoconstriction

To investigate whether the inhibitory effect of Rb_1_ on hyper-vasoconstriction under HG and ED conditions was associated with the activation of the K^+^ channel, the aortic rings were pretreated with K^+^ channel blockers. Briefly, the aortic rings with chemically denuded epithelium were treated with 10^−3^ M TEA to block the Ca^2+^-activated K^+^ channel (K_Ca2+_), 10^−6^ M Gliben to block the K_ATP_, 10^−3^ M BaCl_2_ to block the inward rectifier K^+^ channel (K_IR_), or 10^−3^ M 4-AP to block the voltage-dependent K^+^ channel (K_V_) [20]. The rings were subsequently treated with 10^−5^ M Rb_1_ prior to agonist-induced vasoconstriction.

### 4.4. Measurement of SR Ca^2+^ Release-Induced Vasoconstriction

To determine whether Rb_1_ was involved in PE-induced SR Ca^2+^ release, which causes the exposure of myosin-binding sites on actin and vascular smooth muscle contraction [35], the aortic rings were immersed in Ca^2+^-free Krebs solution supplemented with 10^−4^ M L-NAME, followed by the addition of 10^−5^ M PE to induce vasoconstriction due to SR Ca^2+^ release. The aortas were rinsed with Krebs solution three times to restore lost intracellular Ca^2+^, followed by rinsing with Ca^2+^-free Krebs solution two times. The rings were subsequently pretreated with 10^−5^ M Rb_1_ for 10 min, followed by the addition of 10^−5^ M PE. PE-induced vasoconstriction was calculated to estimate the amount of Ca^2+^ release from the SR [35].

### 4.5. Measurement of Extracellular Ca^2+^ Influx-Induced Vasoconstriction

Two types of Ca^2+^ channels, ROCC and VGCC, were selected to determine whether the effects of Rb_1_ included extracellular Ca^2+^ influx, thereby affecting vasoconstriction under HG and ED conditions. The aortic rings were immersed in Ca^2+^-free Krebs solution containing L-NAME, followed by the addition of 10^−5^ M PE (for Ca^2+^ influx via the ROCC) or 60 mM KCl (for Ca^2+^ influx via the VGCC) to induce vasoconstriction. The rings were subsequently treated with 10^−6^–10^−2^ M CaCl_2_ to obtain concentration–response curves. After rinsing with Ca^2+^-free Krebs solution, to evaluate the inhibitory effect of Rb_1_ on hyper-vasoconstriction induced by HG and ED via the ROCCs or VGCCs, the aortic rings were treated with 10^−5^ M Rb_1_ for 10 min prior to the addition of 10^−5^ M PE or 60 mM KCl and subsequent 10^−6^–10^−2^ M CaCl_2_. The percent PE- or KCl-induced vasoconstriction was calculated based on the maximal contractile response to 10^−2^ M CaCl_2_ [35].

### 4.6. Chemicals

All chemical reagents, including ACh, BaCl_2_, 4-AP, Gliben, glucose, L-NAME, mannitol, NAC, PE, Rb_1_ and Rg_1_, TEA, and dimethylsulfoxide (DMSO), were purchased from Sigma-Aldrich (St. Louis, MO, USA). Rb_1_, Rg_1_, and Gliben were dissolved in DMSO, whereas all other reagents were dissolved in distilled water.

### 4.7. Statistical Analysis

All data were expressed as mean ± SEM and analyzed using SPSS statistics version 26 software (IBM, IL, USA). The data for extracellular Ca^2+^ influx- and SR Ca^2+^ release-induced vasoconstriction were compared by paired *t*-tests. The other data were analyzed using one-way analysis of variance (ANOVA) followed by the least significant difference (LSD) test for post hoc comparison. *p* < 0.05 was deemed statistically significant.

## 5. Conclusions

In this study, we have demonstrated that Rb_1_ effectively reduced hyper-vasoconstriction induced by HG and ED (*p* = 0.004) via the inhibition of Ca^2+^ influx through the ROCC (10^−6^–10^−4^ M CaCl_2_, *p* < 0.001; 10^−3^–2.5 × 10^−3^ M CaCl_2_, *p* = 0.001) and the VGCC (10^−6^ M CaCl_2_, *p* = 0.003; 10^−5^–10^−2^ M CaCl_2_, *p* < 0.001) and partially through the activation of the K_ATP_ in rat aorta for the first time (Figure 3). Our research provides further evidence into clinical applicability that Rb_1_ could be recommended as a therapeutic agent for ED in diabetes and provides a guideline for clinical studies evaluating the importance of managing vascular function in patients with hyperglycemia and ED, including in patients with acute pancreatitis.

## Figures and Tables

**Figure 1 pharmaceuticals-16-01238-f001:**
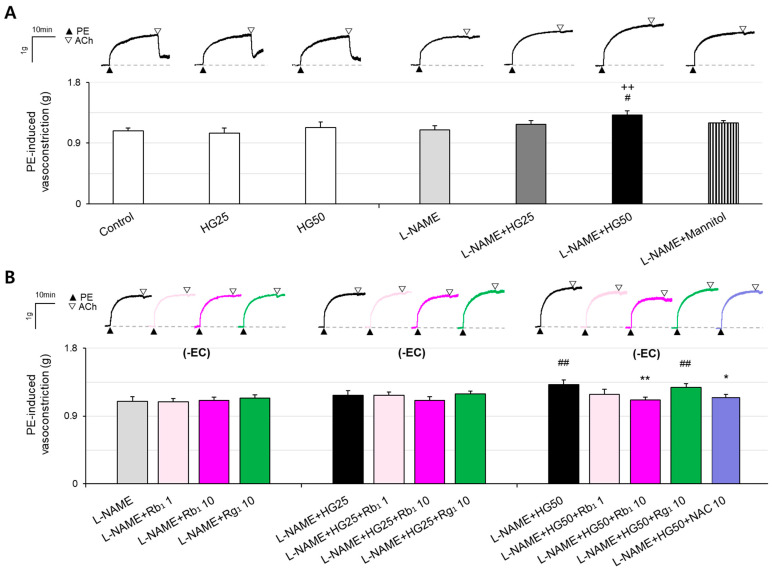
Effect of Rb_1_ on vasoconstriction induced by HG and ED. (**A**) PE (10^−5^ M)-induced vasoconstriction in the presence of HG. (**B**) Effects of ginsenoside Rb1 (10^−6^ and 10^−5^ M) and Rg1 (10^−5^ M) on vascular smooth muscle constriction. Results are presented as the mean ± SEM. Data were analyzed by one-way ANOVA followed by the LSD test for post hoc analysis (*n* = 12–18; ^++^ *p* < 0.01 vs. Control; ^#^
*p* < 0.05 vs. L-NAME; ^##^
*p* < 0.01 vs. L-NAME; * *p* < 0.05 vs. L-NAME + HG50; ** *p* < 0.01 vs. L-NAME + HG50). Abbreviations: ACh, acetylcholine; ED, endothelial dysfunction; HG, high glucose; L-NAME, Nω-Nitro-L-arginine methyl ester hydrochloride; -EC, endothelium-denuded; NAC, N-acetylcysteine; PE, phenylephrine; Rb_1_, ginsenoside Rb1; Rg_1_, ginsenoside Rg1.

**Figure 2 pharmaceuticals-16-01238-f002:**
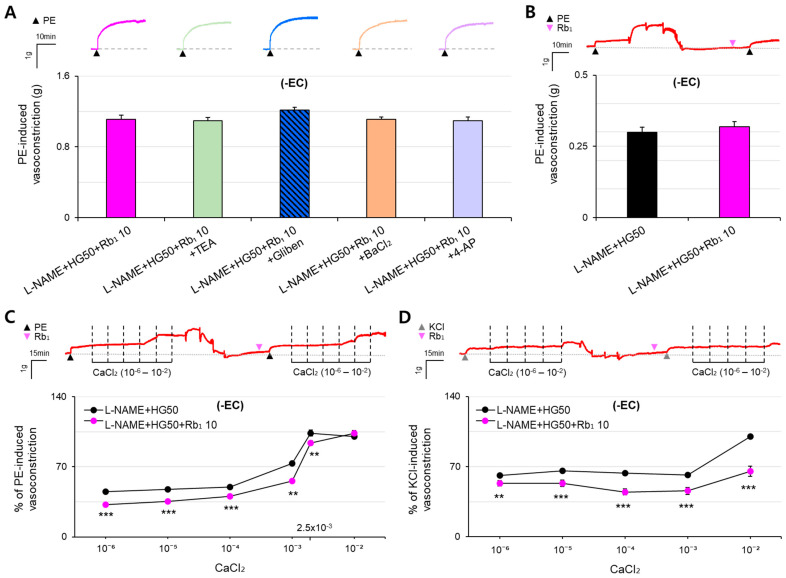
Mechanisms underlying the action of Rb_1_ on hyper-vasoconstriction induced by HG and ED. (**A**) Effect of Rb_1_ (10^−5^ M) on PE (10^−5^ M)-induced vasoconstriction through the K^+^ channel. (**B**) Effect of Rb_1_ (10^−5^ M) on SR Ca^2+^ release-induced vasoconstriction. (**C**,**D**) Effects of Rb_1_ (10^−5^ M) on extracellular Ca^2+^ influx (10^−6^–10^−2^ M CaCl_2_)-induced vasoconstriction via (**C**) ROCC and (**D**) VGCC. All results are presented as the mean ± SEM. Extracellular Ca^2+^ influx and SR Ca^2+^ release were compared by paired *t*-tests (*n* = 12–16; ** *p* < 0.01 vs. L-NAME + HG50; *** *p* < 0.001 vs. L-NAME + HG50). All other parameters were compared by one-way ANOVA followed by LSD tests (*n* = 11–18). Abbreviations: BaCl_2_, barium chloride; ED, endothelial dysfunction; 4-AP, 4-aminopyridine; Gliben, glibenclamide; HG, high glucose; L-NAME, N^G^-nitro-L-arginine methyl ester; -EC, endothelium-denuded; PE, phenylephrine; Rb_1_, ginsenoside Rb1; ROCC, receptor-operated calcium channel; SR, sarcoplasmic reticulum; TEA, tetraethylamine; VGCC, voltage-gated calcium channel.

**Figure 3 pharmaceuticals-16-01238-f003:**
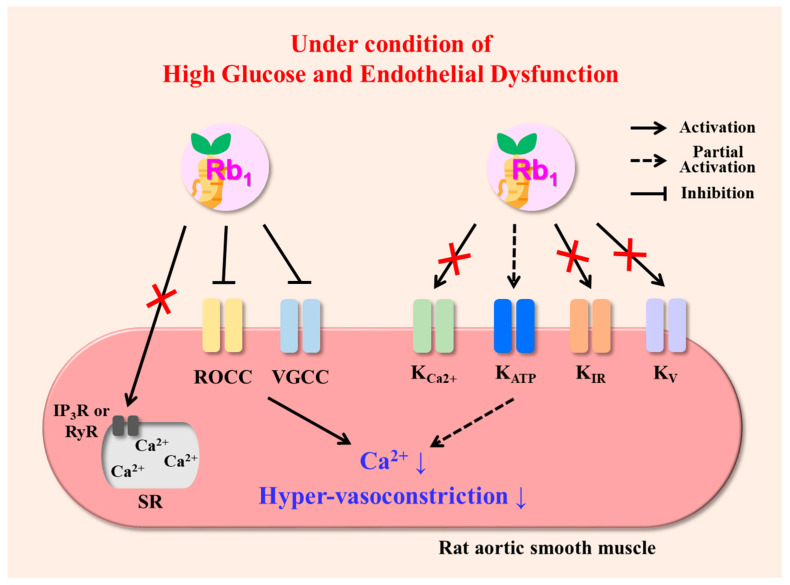
Proposed mechanisms of action of Rb_1_ in reducing hyper-vasoconstriction induced by high glucose and endothelial dysfunction in rat aorta. Abbreviations: IP_3_R, inositol 1,4,5-trisphosphate receptor; K_ATP_, ATP-sensitive K^+^ channel; K_Ca2+_, Ca^2+^-activated K^+^ channel; K_IR_, inward rectifier K^+^ channel; K_V_, voltage-dependent K^+^ channel; Rb_1_, ginsenoside Rb1; ROCC, receptor-operated calcium channel; SR, sarcoplasmic reticulum; VGCC, voltage-gated calcium channel.

## Data Availability

Data are available on request from the corresponding author on reasonable request.

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
