# Peer review of "Ginsenoside Rb1 Reduces Hyper-Vasoconstriction Induced by High Glucose and Endothelial Dysfunction in Rat Aorta"

_pharmaceuticals, 2023, doi:10.3390/ph16091238_

Round 1

Reviewer 1 Report

This is well-orchestrated research to further confirm the beneficial effect of Ginsenoside Rb1 in diabetes. However, characterization of Ginsenoside Rb1 should have also been included in the discussion section and thoroughly discussed. 

OK

Reviewer 2 Report

The “communication” presented for review concerns the effect of Ginsenoside Rb1 on the reduction of hypercontraction and endothelial dysfunction in vascular smooth muscle caused by high glucose levels. This report provides interesting information that Rb1 may have a potential therapeutic application in the treatment of endothelial dysfunction in diabetes. This “communication”, if it is to be published in Pharmaceuticals Journal, requires some revisions. Below are the main comments.

1. In the title it is worth mentioning  that the conducted research concerns the smooth muscles of the aorta in rats

2.  In the “Abstract” section:

·       Please provide the key (quantitative) results of the research carried out, at the moment only general information and trends are given

3.       In the "Introduction" section:

·       The botanical taxonomy and characteristics of Red ginseng (Panax ginseng C. A. Meyer) should be detailed

·       Lines 50-52: It would be worth making at least a brief characterization of ginserosides, since their division into protopanaxadiols / protopanaxatriols has already been mentioned. Main properties, mechanisms of action? Are there any differences or similarities in the physiological effects of these compounds?

·       Lines 62-64: The purpose of the presented research should be clarified, what the Authors mean by using the term "aortic rats" - because it does not result from the purpose, but only from the analysis of the further part of the article. Specify the purpose of the research and emphasize its innovativeness.

4.       In the “Results”and “Discussion”section:

·       It is suggested that in Figure 1 and Figure 2 statistically significant differences should be marked with letter symbols (not with graphic symbols), then their interpretation will be more transparent for the potential reader.

5.       In the "Materials and Methods" section:

·       Subsection “4.1. Animals” – Could you please give some more details on this study, for example the number of animals used in the study? The number of animal groups and the number of animals in each group. What type of diet was used? The term "regular diet" is not enough.

·       It would be worth adding a graphical diagram of the whole experiment

6.       In the "Conclusions" section:  

·       Please add quantitative results

·    The novelty and applicability of the obtained research results should be emphasized.

Reviewer 3 Report

1.      Several grammars and typing errors were noticed in the text. I recommend the authors to carefully proofread the manuscript to rectify these errors.

Including, but not limited to, the following:

Line 59, 3 should be three.

Line 24, channels should be channel.

2.      It is necessary to add relevant detail of parts 4.4 and 4.5.

3.      The research instrument is too homogenous and it is recommended that more research instruments be added to support the current results. Thus, making the conclusions more rigorous.

In summary, this manuscript is not suitable for publication now.

It is OK.

Round 2

Reviewer 2 Report

After reading the revised version of the manuscript corrected by the Authors,
in my opinion, it can be considered for futher stages of publication.
